# Evaluation of automated specialty palliative care in the intensive care unit: A retrospective cohort study

**Katharine E. Secunda**[1]☯*, **Kristyn A. Krolikowski**[2], **Madeline F. Savage**[2], **Jacqueline M. Kruser**[3]☯

**1** Department of Medicine, Division of Pulmonary, Allergy, and Critical Care, Perelman School of Medicine of the University of Pennsylvania, Philadelphia, PA, United States of America, **2** Northwestern University Feinberg School of Medicine, Chicago, IL, United States of America, **3** Department of Medicine, Division of Allergy, Pulmonary and Critical Care, University of Wisconsin School of Medicine and Public Health, Madison, WI, United States of America

☯ These authors contributed equally to this work.
* Katharine.Secunda@pennmedicine.upenn.edu

**Data Availability Statement:** The raw data from this study cannot be shared publicly because they contain potentially identifying and sensitive patient information derived from electronic health records.

## Abstract

### Introduction

Automated specialty palliative care consultation (SPC) has been proposed as an intervention to improve patient-centered care in the intensive care unit (ICU). Existing automated SPC trigger criteria are designed to identify patients at highest risk of in-hospital death. We sought to evaluate common mortality-based SPC triggers and determine whether these triggers reflect actual use of SPC consultation. We additionally aimed to characterize the population of patients who receive SPC without meeting mortality-based triggers.

### Methods

We conducted a retrospective cohort study of all adult ICU admissions from 2012–2017 at an academic medical center with five subspecialty ICUs to determine the sensitivity and specificity of the five most common SPC triggers for predicting receipt of SPC. Among ICU admissions receiving SPC, we assessed differences in patients who met any SPC trigger compared to those who met none.

### Results

Of 48,744 eligible admissions, 1,965 (4.03%) received SPC; 979 (49.82%) of consultations met at least 1 trigger. The sensitivity and specificity for any trigger predicting SPC was 49.82% and 79.61%, respectively. Patients who met no triggers but received SPC were younger (62.71 years vs 66.58 years, mean difference (MD) 3.87 years (95% confidence interval (CI) 2.44–5.30) p<0.001), had longer ICU length of stay (11.43 days vs 8.42 days, MD -3.01 days (95% CI -4.30 – -1.72) p<0.001), and had a lower rate of in-hospital death (48.68% vs 58.12%, p<0.001).

Data requests by researchers who meet criteria for access to confidential data can be directed to the Northwestern University Institutional Review Board (contact via email at irb@northwestern.edu).

**Funding:** K.E.S was supported through Northwestern University Center for Bioethics and Medical Humanities Exploratory Grant. J.M.K was supported, in part, through NIH/NHLBI K23HL146890. The funders had no role in study design, data collection and analysis, decision to publish or preparation of the manuscript.

**Competing interests:** The authors have declared no competing interests exist.

## Conclusion

Mortality-based triggers for specialty palliative care poorly reflect actual use of SPC in the ICU. Reliance on such triggers may unintentionally overlook an important population of patients with clinician-identified palliative care needs.

## Introduction

The delivery of high-quality palliative care in the intensive care unit (ICU) is a cornerstone of critical care medicine and is widely recognized as necessary for the provision of patient- and family-centered ICU care [1–4]. The World Health Organization defines palliative care as an approach to healthcare that improves quality of life for patients with serious illness and their families by attending to physical, psychosocial, and spiritual needs [5]. The delivery of palliative care for patients with critical illness has been associated with increased rates of advanced directives, decreased ICU length of stay, increased use of hospice, and decreased use of non-beneficial life-sustaining therapies in observational studies [6–8]. Despite these benefits, the provision of palliative care in the ICU remains suboptimal [9].

Palliative care in the ICU includes both *primary* palliative care, which is provided by any clinician in the ICU, and *specialty* palliative care consultation (SPC), which entails consultation by palliative care specialists with advanced training. One-quarter of all hospital-based specialty palliative care consultations occur in the ICU [10]. Recent guidelines recommend an integrative palliative care approach wherein a patient receives palliative care concurrently with restorative/curative care from the time of ICU admission in an individualized manner [2]. However, the optimal role of SPC and the relationship between primary and specialty palliative care in the ICU remain poorly defined with few empiric data to guide clinicians' decision making.

Recent efforts to improve palliative care for patients admitted to the ICU have focused on the use of screening criteria or "triggers" to prompt or automate SPC. An analysis of a national database estimated that 14–20% of ICU patients meet at least one of the commonly used, recommended triggers for SPC [11]. Because this need for palliative care cannot be met by the existing palliative care workforce in the United States [12], it is essential to define the optimal role of SPC and primary palliative care in the ICU. Existing triggers have been designed with the primary goal of identifying patients at highest risk of in-hospital death [6, 13–15]. However, it is unknown whether patients at the highest risk for imminent death are the most likely to derive benefit from SPC [16] and, alternatively, whether a mortality-based trigger strategy unintentionally overlooks an important population of patients who may benefit from SPC.

In this study, we sought to evaluate the most common and recommended mortality-based SPC triggers and determine whether these triggers reflect actual use of SPC in a large, urban, academic medical center with multiple subspecialty ICUs. In addition, we aimed to characterize the population of patients who receive SPC guided by clinician-identified need, without meeting mortality-based triggers.

## Methods

### Study design, setting, and participants

We conducted a retrospective cohort study of all adult (age > 18) ICU admissions from January 1st, 2012 to December 31st, 2017 at an 894-bed, urban academic medical center with 115 intensive care beds distributed over five subspecialty ICUs (Surgical Intensive Care Unit

(SICU), Medical Intensive Care Unit (MICU), Cardiac Care Unit (CCU), Neurosciences-spine Intensive Care Unit (NSICU), and a Cardiothoracic/Transplant Intensive Care Unit (CTICU)). No hospital or unit protocols to standardize SPC consultation were used during the study period, apart from routine SPC consultations during evaluation for a cardiac Ventricular Assist Device (VAD). We excluded admissions with an ICU length of stay less than four hours and patients who received SPC consultation prior to the ICU admission during the study period. Deidentified patient-level data was obtained from the Northwestern Medicine Enterprise Data Warehouse, an integrated repository of all electronic health record data from the study site. The Northwestern University Institutional Review Board reviewed and approved the study, and a waiver of informed consent was obtained.

## Variables

Admissions were defined as receiving SPC if an initial palliative care consultation report was documented in the electronic health record (EHR) during the ICU admission. Admissions were defined as having a SPC trigger if they met at least one of the five most recommended, published PC triggers: (1) ICU admission following a hospital stay greater than or equal to 10 days; (2) age greater than 80 with two or more life-threatening comorbidities (as defined by the Acute Physiology and Chronic Health Evaluation IV definitions of severe chronic organ insufficiency); (3) diagnosis of active stage IV malignancy; (4) status post cardiac arrest; or (5) diagnosis of intracerebral hemorrhage requiring mechanical ventilation [7, 11, 16].

The reasons for SPC were abstracted from the first palliative care consultation report in the EHR from the "Reason for Consult" or "Reason for Encounter" section of the note template. Reason for consult is selected by the note writer from a drop-down box within the note template; selection from this list is not mandatory for consult completion. If a note did not include information within this section, the entire content of the note was reviewed for additional free-text statements ("palliative care consulted for [. . .]" or "asked to see patient to [. . .]") that explicitly stated the reasons for consultation. Two investigators independently reviewed and categorized all non-template, free-text consult reasons, and a third investigator arbitrated all discrepancies. If an SPC note included more than one reason for consultation, all reasons were abstracted and included in the analysis. If an SPC did not contain any stated reason for consultation, the encounter was designated as "no evident reason for consultation", which was categorized as "other" during analyses. Reasons for consultation were grouped into nine categories based on the palliative care consult note template options: (1) goals of care, (2) symptom management, (3) end-of-life care management, (4) disposition planning, (5) medical decision making, (6) patient/family support, (7) advance care planning, (8) consideration of mechanical circulatory support, and (9) other, including no evident reason for consultation.

## Statistical analysis

We assessed differences in demographics, clinical characteristics, and outcomes between the ICU admissions that received a new specialty palliative care consultation and those that did not, using t-tests and chi-square tests of independence. We repeated these analyses for ICU admissions that had received SPC, comparing those who had at least one SPC trigger with those that did not meet any trigger criteria. Of ICU admissions with a palliative care consultation, we conducted t-tests to evaluate for any differences in the reasons for SPC, again between those who had at least one SPC trigger and those without.

We evaluated the accuracy of using SPC trigger criteria to predict receipt of SPC when compared to common illness severity scores associated with in-hospital mortality. For all ICU admissions, a Receiver-Operating-Characteristic (ROC) curve analysis was conducted and

area under the curve (AUC) was calculated to determine how accurately the Sepsis-related Organ Failure Assessment (SOFA) score [17–19] and Acute Physiology Score (APS) [20] at the time of ICU admission predicted new specialty palliative care consultation in the ICU. The calculated Youden index was used to determine cut-off values for the SOFA and APS scores that resulted in optimal sensitivity and specificity. Sensitivity and specificity for predicting actual receipt of SPC were calculated for: (1) one or more trigger criteria, (2) each individual trigger criterion, (3) admission APS score, and (4) admission SOFA score. Statistical significance was defined by a 2-tailed P-value of <0.05. SAS version 9.4 software (Cary, NC, USA) was used for all analyses [21].

## Results

Of 48,744 eligible ICU admissions, 10,513 (21.57%) met one or more triggers for SPC, and 1,965 (4.03%) received specialty palliative care consultation (Fig 1). The trigger most frequently met was a diagnosis of active stage IV malignancy (38.88%), followed by ICU admission after a hospital stay greater than 10 days (27.36%), age greater than 80 years with two or more life-threatening comorbidities (24.26%), ICU admission after cardiac arrest (15.45%),

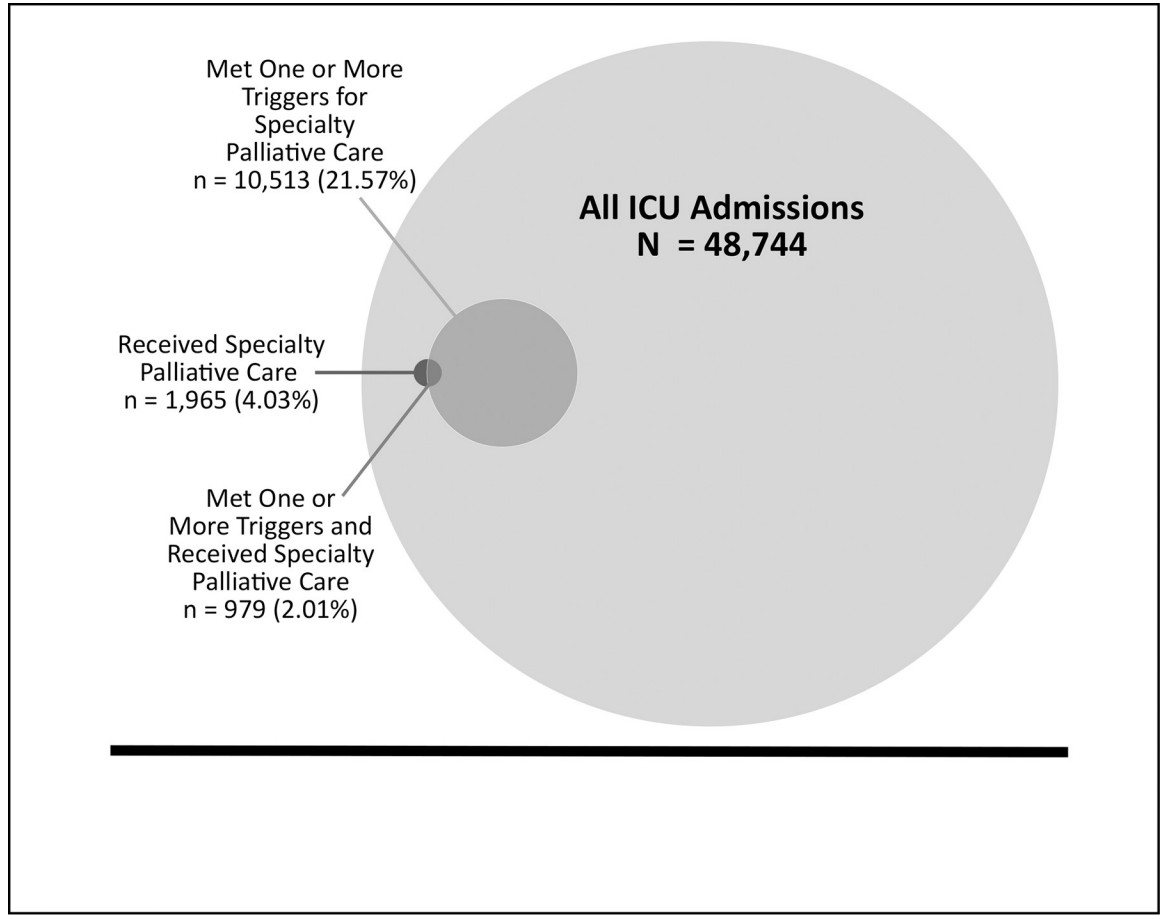

**Fig 1. The relationship between actual specialty palliative care delivery and recommended trigger criteria among all adults admitted to the intensive care unit.** Of 48,744 eligible intensive care unit admissions, 10,513 (21.57%) met one or more triggers for specialty palliative care consultations, and 1,965 (4.03%) received specialty palliative care consultation. Of all intensive care unit admissions, only 979 (2.01%) both received specialty palliative care and met one or more triggers for specialty palliative care.

and intracerebral hemorrhage requiring mechanical ventilation (5.10%). The rate of SPC by subspecialty ICU was: 1,018 (6.81%) of MICU admissions, 292 (4.77%) of CCU admissions, 319 (2.83%) of NSICU admissions, 205 (2.24%) of CTICU admissions, and 131 (1.81%) of SICU admissions (p <0.001). The table in S1 Table shows the demographics, clinical characteristics, and hospitalization outcomes of the entire cohort, comparing those who received SPC to those who did not.

Of the 1,965 ICU admissions that included specialty palliative care consultation, 979 (49.82%) met one or more SPC triggers (Table 1). Admissions that did not meet any SPC triggers but received SPC were younger than admissions that met triggers (62.71 years vs 66.58 years, mean difference (MD) 3.87 years (95% CI 2.44–5.30), p<0.0001). Illness severity by SOFA or APS score did not differ between groups. Admissions that did not meet any triggers had more days from ICU admission to specialty palliative care consultation (6.63 days vs 5.20 days, MD -1.43 days (95% CI -2.24 – -0.62), p<0.001) and longer ICU length of stay (11.43 days vs 8.42 days, MD -3.01 days (95% CI -4.30 – -1.72), p < 0.001) compared to those that met triggers. In the CCU and CTICU, there were more trigger-negative compared to trigger-positive admissions receiving SPC (18.97% vs 10.73%, p < 0.001, and 14.30% vs 6.45%, p < 0.001, respectively). Admissions that did not meet any triggers were less likely to die in the hospital than those meeting triggers (48.68% vs 58.12%, p < 0.001). Among survivors, hospital discharge destination differed for trigger-negative vs trigger-positive admissions, respectively: 48.42% vs 43.90% were discharged to home, 20.36% vs 30.00% to hospice, and 28.26% vs 23.90% to a post-acute care facility (p = 0.008).

Among the 1,965 admissions that included SPC, the reasons for consultation differed by trigger status (Table 2). Goals of care, symptom management, and disposition planning were more frequent reasons for consultation in the trigger-positive admissions compared to trigger negative admissions (65.58% vs 57.51%, MD 0.08 (95% CI 0.04–0.12), p<0.001; 39.94% vs 32.76%, MD 0.07 (0.03–0.11), p<0.001; and 18.59% vs 14.60%, MD 0.04 (95% CI 0.01–0.07), p = 0.018, respectively). Automated consultation prior to VAD was more frequent in trigger-negative compared to trigger-positive admissions (9.03% vs 1.43%, MD -0.08 (95% CI -0.10 – -0.06), p < 0.001).

The sensitivity and specificity for any trigger predicting actual specialty palliative care consultation was 49.82% and 79.61%, respectively (Table 3). The APS and SOFA scores at the time of ICU admission had sensitivities of 66.87% and 65.60% and specificities of 64.82% and 63.97%, respectively. The AUCs for any SPC trigger, APS score, and SOFA scores were 0.65, 0.72, and 0.69, respectively (S1 Fig demonstrates the ROC curves for APS score and SOFA score).

## Discussion

In this retrospective cohort study of almost 50,000 critically ill patients from a large academic medical center, we found that commonly recommended, mortality-based triggers for automated SPC are neither sensitive nor specific for predicting actual receipt of specialty palliative care consultation in the ICU. We also identified and characterized a population of younger patients with prolonged critical illness who are overlooked by mortality-based triggers but have a clinician-identified need for SPC. Because the ideal role of SPC in the ICU has not been established, definitive interpretation of the appropriateness of specialty palliative care utilization in this study cannot be determined. Nevertheless, our findings underscore the need to develop a more refined approach to optimize specialty palliative care delivery in the ICU and to overcome the common misperception that specialty palliative care is only relevant for patients facing imminent death.

**Table 1. Demographics, clinical characteristics, and hospitalization outcomes of intensive care unit admissions who received specialty palliative care consultation in the intensive care unit.**

| | All admissions that received specialty palliative care<br>N = 1965 | Admissions with one or more triggers<br>N = 979 | Admissions with no triggers<br>N = 986 | Mean difference or df, N | P-value |
|---|---|---|---|---|---|
| Age in years, mean (95% CI) | 64.64 (63.92–65.36) | 66.58 (65.55–67.60) | 62.71 (61.71–63.71) | 3.87 (2.44–5.30) | <0.001[a] |
| Female, No./total No. (%) | 918/1965 (46.72) | 473/979 (48.31) | 445/986 (45.13) | df = 1, N = 1965 | 0.157[b] |
| Race, No./total No. (%) | | | | df = 4, N = 1965 | 0.167[b] |
| White or Caucasian | 1031/1965 (52.47) | 529/979 (54.03) | 502/986 (50.91) | | - |
| Black or African American | 399/1965 (20.31) | 186/979 (19.00) | 213/986 (21.60) | | - |
| Asian | 67/1965 (3.41) | 40/979 (4.09) | 27/986 (2.74) | | - |
| Other Race[c] | 215/1965 (10.94) | 107/979 (10.93) | 108/986 (10.95) | | - |
| Unknown Race[d] | 253/1965 (12.88) | 117/979 (11.95) | 136/986 (13.79) | | |
| Hispanic or Latinx Ethnicity | 145/1965 (7.38) | 70/979 (7.15) | 75/986 (7.61) | df = 2, N = 1965 | 0.656[b] |
| Illness severity, mean (95% CI) | | | | | |
| APS Score | 55.27 (54.07–56.46) | 56.26 (54.54–57.99) | 54.27 (52.62–55.92) | 2.00 (-0.39–4.39) | 0.101[a] |
| SOFA score | 5.69 (5.51–5.86) | 5.70 (5.45–5.94) | 5.68 (5.43–5.93) | 0.01 (-0.34–0.36) | 0.939[a] |
| ICU type, No./total No. (%) | | | | df = 4, N = 1965 | <0.001[b] |
| MICU | 1018/1965 (51.81) | 558/979 (57.00) | 460/986 (46.65) | | - |
| NSICU | 319/1965 (16.23) | 185/979 (18.90) | 134/986 (13.59) | | - |
| CCU | 205/1965 (10.43) | 105/979 (10.73) | 187/986 (18.97) | | - |
| CTICU | 131/1965 (6.67) | 64/979 (6.54) | 141/986 (14.30) | | - |
| SICU | 292/1965 (14.86) | 67/979 (6.84) | 64/986 (6.49) | | - |
| ICU length of stay in days, mean (95% CI) | 9.93 (9.29–10.57) | 8.42 (7.67–9.16) | 11.43 (10.38–12.47) | -3.01 (-4.30 – -1.72) | <0.001[a] |
| In-hospital death, No./total No. (%) | 1049/1965 (53.38) | 569/979 (58.12) | 480/986 (48.68) | df = 1, N = 1965 | <0.001[b] |
| Discharge disposition among survivors[e] | | | | df = 3, N = 914 | 0.008[b] |
| Home | 425/914 (46.40) | 180/409 (43.90) | 245/505 (48.42) | | - |
| Hospice[f] | 226/914 (24.67) | 123/409 (30.00) | 103/505 (20.36) | | - |
| Post-acute Care Facility[g] | 212/914 (26.31) | 98/409 (23.90) | 143/505 (28.26) | | - |
| Other[h] | 22/914 (2.40) | 8/409 (1.95) | 14/505 (2.77) | | - |

[a] T-test.

[b] $\chi^2$ test of independence.

[c] Other Race includes American Indian or Alaska Native, Hispanic or Latinx, Native Hawaiian or other Pacific Islander, or Other.

[d] Unknown Race includes Declined, Unable to answer, Unknown, or missing data.

[e] Statistics were calculated using available data.

[f] Hospice includes Expiration Hospice, Home with Hospice, or Inpatient Hospice.

[g] Post-acute Care Facility includes Acute Inpatient Rehabilitation, Long-term Acute Care hospital (LTAC), Nursing Home (Custodial), and Skilled nursing facility or subacute rehab.

[h] Other includes Acute Care Hospital, Against Medical Advice (AMA) or Elopement, Intermediate care facility including state, Other, or VA System Facility.

Abbreviations: APS = Acute Physiology Score, CCU = Cardiac Care Unit, CTICU = Cardiothoracic/Transplant Intensive Care Unit, ICU = Intensive Care Unit, MICU = Medical Intensive Care Unit, NSICU = Neurosciences-spine Intensive Care Unit, SICU = Surgical Intensive Care Unit, SOFA = Sequential Organ Failure Assessment, SPC = Specialty Palliative Care.

The findings of this study build on and align with recent work that demonstrates the limitations of existing SPC triggers and seeks to define the optimal relationship between primary and specialty palliative care in the ICU. Hua and colleagues have focused on the need to develop novel triggers for SPC that are based on risk of longer-term outcomes instead of in-

**Table 2. Characteristics of specialty palliative care consultations in the intensive care unit.**

| | All admissions that received specialty palliative care | Admissions with one or more triggers | Admissions with no triggers | Mean difference | P-value[a] |
|---|---|---|---|---|---|
| | N = 1965 | N = 979 | N = 986 | | |
| Days from ICU admission to consult, mean (95% CI) | 5.92 (5.51–6.32) | 5.20 (4.70–5.70) | 6.63 (5.99–7.26) | -1.43 (-2.24 – -0.62) | <0.001 |
| Days from consult to hospital discharge, mean (95% CI) | 4.01 (3.57–4.45) | 3.22 (2.79–3.65) | 4.80 (4.04–5.55) | -1.58 (-2.45 – -0.71) | <0.001 |
| Reasons for consultation, No./Total No. (%) | | | | | |
| Goals of care | 1209/1965 (61.53) | 642/979 (65.58) | 567/986 (57.51) | 0.08 (0.04–0.12) | <0.001 |
| Symptom management | 714/1965 (36.34) | 391/979 (39.94) | 323/986 (32.76) | 0.07 (0.03–0.11) | <0.001 |
| End-of-life management | 413/1965 (21.02) | 221/979 (22.57) | 192/986 (19.47) | 0.03 (-0.01–0.07) | 0.092 |
| Disposition planning | 326/1965 (16.59) | 182/979 (18.59) | 144/986 (14.60) | 0.04 (0.01–0.07) | 0.018 |
| Automated consult prior to VAD/cardiac mechanical support | 103/1965 (5.24) | 14/979 (1.43) | 89/986 (9.03) | -0.08 (-0.10 – -0.06) | <0.001 |
| Other | 101/1965 (5.14) | 43/979 (4.39) | 58/986 (5.88) | -0.01 (-0.03–0.01) | 0.135 |
| Patient/family Support | 46/1965 (2.34) | 24/979 (2.45) | 22/986 (2.23) | 0.00 (-0.01–0.02) | 0.747 |
| Medical decision making | 39/1965 (1.98) | 17/979 (1.74) | 22/986 (2.23) | -0.01 (-0.02–0.01) | 0.432 |
| Advanced care planning | 38/1965 (1.93) | 15/979 (1.53) | 23/986 (2.33) | -0.00 (-0.02–0.00) | 0.198 |

[a] T-test.

Abbreviations: VAD = Ventricular Assist Device.

hospital mortality and have demonstrated that existing SPC triggers have low sensitivity and high specificity for predicting 6-month mortality [22]. In a large, multicenter survey of ICU clinicians, the vast majority of participants felt that that SPC was underutilized in the ICU and endorsed an automated-trigger based system to protocolize consultation [9]. While clinicians approved of the existing approach to SPC triggers that reflect a high risk of in-hospital

**Table 3. Accuracy of mortality-based triggers and severity of illness scores for predicting specialty palliative care consultation in the intensive care unit.**

| | Sensitivity | Specificity | AUC |
|---|---|---|---|
| APS score at time of ICU admission[a] | 66.87% | 64.82% | 0.72 |
| SOFA score at time of ICU admission[a] | 65.60% | 63.97% | 0.69 |
| One or more triggers met | 49.82% | 79.61% | 0.65 |
| Individual triggers: | | | |
| Metastatic cancer | 25.39% | 92.33% | 0.59 |
| Greater than 10 day hospital stay before ICU admission | 13.54% | 94.42% | 0.54 |
| Age greater than 80 plus two or more life threatening conditions | 10.28% | 94.98% | 0.53 |
| ICU admission following cardiac arrest | 6.26% | 96.79% | 0.52 |
| Intracerebral hemorrhage requiring mechanical ventilation | 3.82% | 99.01% | 0.51 |

[a] The calculated Youden index was used to determine cut-off values for the SOFA and APS scores that resulted in optimal sensitivity and specificity. A cut-off of 39.07 was used for APS score and a cut-off of 4.00 was used for SOFA score.

Abbreviations: APS = Acute Physiology Score, AUC = Area Under the Curve, ICU = Intensive Care Unit, SOFA = Sequential Organ Failure Assessment.

mortality, they also endorsed the importance of non-mortality-based triggers such as unrealistic expectations and clinician-family conflict that have not been, to date, incorporated into SPC trigger criteria [9]. Clinician endorsement of the role of SPC in the ICU beyond the narrow focus of imminent death is consistent with our finding that over half of patients for whom clinicians request SPC consultation do not meet any of the existing mortality-based trigger criteria.

Patients in this study with a clinician-identified need for SPC but who did not meet any of the existing triggers differed in important ways from patients who met triggers. This population of patients are younger, have lengthy ICU admissions, are less likely to die, and more likely to receive SPC late into their ICU stay. The benefits of integrating palliative care early in the course of advanced cancer have been well-described [23, 24], and there is mounting evidence that the timing of in-hospital SPC is important. Early palliative care consultation in the ICU (defined as within 4 days of ICU admission) has been associated with reductions in hospital length of stay and direct costs [25]. A study of hospitalized oncologic patients found that late specialty palliative care consultations (defined as greater than one week into admission) were associated with a marked increase in health utilization outcomes [26]. Prior studies have also demonstrated a high burden of unmet palliative care needs for patients living in long-term care or nursing facilities [27–29], and in this study we found that more than a quarter of trigger-negative patients who receive SPC are discharged to post-acute care facilities. Taken together, our findings that a significant portion of trigger-negative patients receive SPC late in their course and are discharged to post-acute care facilities highlights an opportunity to develop novel triggers that proactively identify this group of patients with SPC needs. Our findings align with ongoing efforts to leverage the electronic health record to support the delivery of specialty palliative care by developing more nuanced predictive models and screening algorithms that move beyond in-patient mortality and combine patient characteristics, patient and family self-reported needs and symptoms, and risk for short- and long-term morbidity and mortality [30–34].

We found that in-hospital mortality-based triggers were particularly insensitive to the SPC needs among patients in the two cardiovascular-focused intensive care units included in this study. This finding may be explained, in part, because existing triggers were developed in medical, surgical, or neurological patient populations [16]. Nevertheless, prior literature demonstrates important unmet palliative care needs in this population; patients with advanced heart failure report similar or higher number of symptoms than patients with advanced cancer [35], are referred later to palliative care [36], and are more likely than those with cancer to be referred from a critical care unit [36]. Despite professional guideline recommendations for palliative care involvement for heart failure patients [37], there is no clear consensus regarding when and how to implement palliative care services. Slavin and Warraich suggest initiating specialty palliative care referral at critical moments in the trajectory of heart failure patients, such as at the time of hospitalization or evaluation for certain procedures [38]. Our work affirms the inadequacy of mortality-based triggers for identifying SPC needs among the population of patients with cardiovascular critical illness.

"Goals of care" was by far the most frequent stated reason for SPC, regardless of trigger status, which is consistent with previous work highlighting the need to establish realistic and patient-identified, overarching goals of medical treatments among hospitalized patients [39, 40]. However, the use and meaning of the specific phrase "goals of care" in clinical care is non-specific [41], and is used in clinical documentation to indicate a variety of constructs unrelated to a patient's individual values and preferences, including poor prognosis, to describe conflict with families, or to provide rationale for limitations on specific medical interventions [42]. The ambiguity and broad range of palliative care constructs associated with this phrase may

partly explain the high frequency of "goals of care" as a reason for consultation and may also indicate that clinicians are challenged to clearly articulate their rationale for requesting the assistance of specialty palliative care consultants in the care of critically ill patients.

Our study has strengths and limitations. The study setting of a large, academic health center with five separate ICUs and a high volume of critically ill patients without standardized specialty palliative care consultation provides the opportunity to study and uncover clinician-derived practice patterns. However, hospital-level variability in specialty palliative care availability and practice patterns is well-described [43], thus our results from a single center may not be generalizable to other institutions who have differing palliative care resources, patient populations, or care delivery models. Future work could expand on our findings by exploring the use of SPC triggers across multiple medical centers. Because we were unable to measure whether patients' palliative care needs were truly met during their ICU stays through either primary or specialty palliative care, we are unable to determine the appropriateness (including underutilization or overutilization) of SPC in this cohort or whether SPC was an effective mechanism to meet these needs. The documented reason for consultation in the EHR may not fully capture the intended purpose of the consultation and may not reflect all of the palliative care needs ultimately identified by the consultant. Future study using qualitative methods may provide a more rich and nuanced understanding of what motivates ICU clinicians to request SPC.

## Conclusions

Mortality-based triggers designed to support automated specialty palliative care consultation in the ICU poorly reflect actual delivery of SPC guided by clinician-identified needs. In this study, patients who received SPC but did not meet any existing SPC triggers were younger, had longer ICU length of stay, were less likely to die in the hospital, and received SPC later in their course of illness. Reliance on automated, mortality-based triggers to improve the delivery of palliative care in the ICU may unintentionally overlook or delay SPC in an important population of patients with specialty palliative care needs but who are at lower risk for in-hospital mortality. To achieve optimal palliative care delivery in the ICU, system-level interventions such as automated triggers should focus on the broader role of SPC for patients with serious illness, beyond the narrow focus of imminent end of life.

## Supporting information

**S1 Fig. Receiver Operating Characteristic curves for APS and SOFA severity scores for predicting specialty palliative care consultation among all ICU admissions.** Abbreviations: APS = Acute Physiology Score, SOFA = Sequential Organ Failure Assessment.
(TIF)

**S1 Table. Demographics, clinical characteristics, and hospitalization outcomes of all intensive care unit admissions, stratified by receipt of specialty palliative care consultation.** [a] T-test. [b] $\chi^2$ test of independence. [c] Statistics were calculated using available data. [d] Other Race includes American Indian or Alaska Native, Hispanic or Latinx, Native Hawaiian or other Pacific Islander, or Other. [e] Unknown Race includes Declined, Unable to answer, Unknown, or missing data. Abbreviations: APS = Acute Physiology Score, CCU = Cardiac Care Unit, CTICU = Cardiothoracic/Transplant Intensive Care Unit, ICU = Intensive Care Unit, MICU = Medical Intensive Care Unit, NSICU = Neurosciences-spine Intensive Care Unit, SICU = Surgical Intensive Care Unit, SOFA = Sequential Organ Failure Assessment.
(DOCX)

## Acknowledgments

Guarantor statement: K.E.S had full access to all data in the study and takes responsibility for the integrity of the data and the accuracy of the analysis.

## Author Contributions

**Conceptualization:** Katharine E. Secunda, Jacqueline M. Kruser.

**Data curation:** Katharine E. Secunda, Kristyn A. Krolikowski, Jacqueline M. Kruser.

**Formal analysis:** Katharine E. Secunda, Kristyn A. Krolikowski, Madeline F. Savage, Jacqueline M. Kruser.

**Funding acquisition:** Katharine E. Secunda, Jacqueline M. Kruser.

**Methodology:** Katharine E. Secunda, Kristyn A. Krolikowski, Jacqueline M. Kruser.

**Project administration:** Jacqueline M. Kruser.

**Software:** Kristyn A. Krolikowski.

**Supervision:** Jacqueline M. Kruser.

**Validation:** Kristyn A. Krolikowski.

**Writing – original draft:** Katharine E. Secunda, Jacqueline M. Kruser.

**Writing – review & editing:** Katharine E. Secunda, Jacqueline M. Kruser.

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
