## [Decision Letter · Decision Letter 0]

17 May 2021

PONE-D-21-08904

Evaluation of Automated Specialty Palliative Care in the Intensive Care Unit: A Retrospective Cohort Study

PLOS ONE

Dear Dr. Secunda,

Thank you for submitting your manuscript to PLOS ONE. After careful consideration, we feel that it has merit but does not fully meet PLOS ONE’s publication criteria as it currently stands. Therefore, we invite you to submit a revised version of the manuscript that addresses the points raised during the review process.

We look forward to receiving your revised manuscript.

Kind regards,

Tai-Heng Chen, M.D.

Academic Editor

PLOS ONE

Journal Requirements:

3. We noted in your submission details that a portion of your manuscript may have been presented or published elsewhere.

[A portion of this work was published as an American Thoracic Society International Conference abstract. ]

Please clarify whether this conference proceeding was peer-reviewed and formally published. If this work was previously peer-reviewed and published, in the cover letter please provide the reason that this work does not constitute dual publication and should be included in the current manuscript.

Reviewers' comments:

Reviewer's Responses to Questions

**Comments to the Author**

1. Is the manuscript technically sound, and do the data support the conclusions?

Reviewer #1: Yes

Reviewer #2: Yes

Reviewer #3: Yes

2. Has the statistical analysis been performed appropriately and rigorously? 

Reviewer #1: Yes

Reviewer #2: Yes

Reviewer #3: Yes

3. Have the authors made all data underlying the findings in their manuscript fully available?

Reviewer #1: Yes

Reviewer #2: Yes

Reviewer #3: No

4. Is the manuscript presented in an intelligible fashion and written in standard English?

Reviewer #1: Yes

Reviewer #2: Yes

Reviewer #3: Yes

5. Review Comments to the Author

Reviewer #1: very relevant data.Your study needs to include later other centers furthermore it is ,I think very interesting to study different categories of patients like patients COVID 19 infected as this subject is a big field of research and evaluation

Reviewer #2: The study is well conducted and the conclusions are appropriate. The authors show the need for new triggers for automated speciality palliative care in ICU. Further investigation could be useful in assessing the effectiveness of the implementation with these new triggers.

Reviewer #3: This manuscript by Secunda and coworkers reports the results of a retrospective cohort study in the intensive care unit (ICU) setting. The study aimed to examine whether the most commonly used Specialty Palliative Care (SPC) triggers accurately represent actual SPC consult in the ICU at an academic center. Using SPC triggers alone may miss subsets of patients who may otherwise benefit from SPC (e.g., younger patients). The authors state that the mortality-based SPC triggers used in this study are ones most commonly used and discussed in literature, which include: ICU admission following hospital stay >= 10 days, age >80 with 2 or more life-threatening comorbidities, stage 4 malignancy, post cardiac arrest, and intracerebral hemorrhage requiring mechanical ventilation.

The authors found that these conventional mortality-based criteria for SPC triggers have poor sensitivity and specificity for actual SPC consults. They did acknowledge that they cannot evaluate the appropriateness of the actual SPC utilization due to inability to measure whether the patient’s palliative care need was truly met.

The implications and future directions of this study were well stated in the discussion: to develop a more patient-centered predictive model to better and earlier identify patients who may benefit from SPC.

Minor comments:

Please follow the STROBE guideline for reporting more closely. Use the predefined headers.

The IRB approval statement typically appears in the first paragraph of the methods.

Line 101-102: “Multiple reasons for consultation could be used for single SPC encounter, and some notes did not document an explicit reason for consultation.” How were these data treated?

Table 1: Why is the denominator for race in all 3 columns not corresponding to the total number of subjects? For example, the first column under race shouldn’t the denominator be 1965 instead of 1896? What happened to the missing numbers? Could be due to “unknown” category? If so, would add an “unknown” category for race.

6. PLOS authors have the option to publish the peer review history of their article (what does this mean?). If published, this will include your full peer review and any attached files.

Reviewer #1: **Yes: **Pr Hanane EZZOUINE

Reviewer #2: No

Reviewer #3: No

---

## [Author Response · Author response to Decision Letter 0]

7 Jul 2021

Reviewer comments

Reviewer's Responses to Questions

1. Is the manuscript technically sound, and do the data support the conclusions?

Reviewer #1: Yes

Reviewer #2: Yes

Reviewer #3: Yes

2. Has the statistical analysis been performed appropriately and rigorously?

Reviewer #1: Yes

Reviewer #2: Yes

Reviewer #3: Yes

3. Have the authors made all data underlying the findings in their manuscript fully available?

The PLOS Data policy requires authors to make all data underlying the findings described in their manuscript fully available without restriction, with rare exception (please refer to the Data Availability Statement in the manuscript PDF file). The data should be provided as part of the manuscript or its supporting information or deposited to a public repository. For example, in addition to summary statistics, the data points behind means, medians and variance measures should be available. If there are restrictions on publicly sharing data—e.g. participant privacy or use of data from a third party—those must be specified.

Reviewer #1: Yes

Reviewer #2: Yes

Reviewer #3: No

Author response: The raw data from this study cannot be shared publicly because they contain potentially identifying and sensitive patient information derived from electronic health records. Data requests by researchers who meet criteria for access to confidential data can be directed to the Northwestern University Institutional Review Board (contact via email at irb@northwestern.edu).

4. Is the manuscript presented in an intelligible fashion and written in standard English?

Reviewer #1: Yes

Reviewer #2: Yes

Reviewer #3: Yes

Reviewer Comments to the Author

Reviewer #1: very relevant data. Your study needs to include later other centers furthermore it is I think very interesting to study different categories of patients like patients COVID 19 infected as this subject is a big field of research and evaluation

Author response: Thank you for your comments. We agree that future research should examine the use of specialty palliative care triggers in different patient populations such as those with COVID-19. We also agree that future work should explore the use of specialty palliative care triggers across multiple medical centers. 

Reviewer #2: The study is well-conducted and the conclusions are appropriate. The authors show the need for new triggers for automated specialty palliative care in ICU. Further investigation could be useful in assessing the effectiveness of the implementation with these new triggers.

Author response: Thank you for your comments. We agree that future work should involve development and implementation of more patient-centered triggers to better identify patients who may benefit from specialty palliative care. 

Reviewer #3: 

1. This manuscript by Secunda and coworkers reports the results of a retrospective cohort study in the intensive care unit (ICU) setting. The study aimed to examine whether the most commonly used Specialty Palliative Care (SPC) triggers accurately represent actual SPC consult in the ICU at an academic center. Using SPC triggers alone may miss subsets of patients who may otherwise benefit from SPC (e.g., younger patients). The authors state that the mortality-based SPC triggers used in this study are ones most commonly used and discussed in literature, which include: ICU admission following hospital stay >= 10 days, age >80 with 2 or more life-threatening comorbidities, stage 4 malignancy, post cardiac arrest, and intracerebral hemorrhage requiring mechanical ventilation.

The authors found that these conventional mortality-based criteria for SPC triggers have poor sensitivity and specificity for actual SPC consults. They did acknowledge that they cannot evaluate the appropriateness of the actual SPC utilization due to inability to measure whether the patient’s palliative care need was truly met.

The implications and future directions of this study were well stated in the discussion: to develop a more patient-centered predictive model to better and earlier identify patients who may benefit from SPC.

2. Minor comments

 a.) Please follow the STROBE guideline for reporting more closely. Use the predefined headers.

 Author response: We appreciate your comment and have updated the manuscript to follow the STROBE guidelines more closely. 

b.) The IRB approval statement typically appears in the first paragraph of the methods.

Author response: We appreciate your comment and have moved the IRB approval statement to the first paragraph in the methods section in the revised manuscript. 

c.) Line 101-102: “Multiple reasons for consultation could be used for single SPC encounter, and some notes did not document an explicit reason for consultation.” How were these data treated?

Author response: We appreciate the opportunity to further clarify how we categorized reasons for specialty palliative care consultation. We have updated the methods section in the revised manuscript with the following clarification statement:

If an SPC note included more than one reason for consultation, all reasons were abstracted and included in the analysis. If an SPC did not contain any stated reason for consultation, the encounter was designated as “no evident reason for consultation”, which was categorized as “other” during analyses.

d.) Table 1: Why is the denominator for race in all 3 columns not corresponding to the total number of subjects? For example, the first column under race shouldn’t the denominator be 1965 instead of 1896? What happened to the missing numbers? Could be due to “unknown” category? If so, would add an “unknown” category for race.

Author response: Thank you for pointing out this additional opportunity to clarify our results. In the dataset used in this study, “Unknown” is a selection made in the electronic medical record for the demographic variables race and ethnicity. “Missing” signifies that data is absent for these variables. We originally treated “unknown” differently than missing data which is why the original denominator was 1896 (which excluded the admissions missing race and/or ethnicity data). However, upon further consideration of these categories based on your comments, we agree that missing race and ethnicity data should be treated the same as “unknown” in a separate category. Thus, we have updated table 1 and the supplementary table to reflect the following new categories for race: (1) White or Caucasian, (2) Black or African American, (3) Asian, (4) Other Race (which includes American Indian or Alaska Native, Hispanic or Latinx Race, Native Hawaiian or other Pacific Islander, and Other), and (5) Unknown Race (which includes declined, unable to answer, unknown, or missing data). We have also updated the statistics to reflect this change which includes the unknown category (which is a combination of the available data stating that race is unknown and missing data). Please see the revised manuscript, Table 1, and Supplemental Table for specific tracked changes.

---

## [Decision Letter · Decision Letter 1]

28 Jul 2021

Evaluation of Automated Specialty Palliative Care in the Intensive Care Unit: A Retrospective Cohort Study

PONE-D-21-08904R1

Dear Dr. Secunda,

We’re pleased to inform you that your manuscript has been judged scientifically suitable for publication and will be formally accepted for publication once it meets all outstanding technical requirements.

Kind regards,

Tai-Heng Chen, M.D.

Academic Editor

PLOS ONE

Reviewers' comments:

Reviewer's Responses to Questions

**Comments to the Author**

1. If the authors have adequately addressed your comments raised in a previous round of review and you feel that this manuscript is now acceptable for publication, you may indicate that here to bypass the “Comments to the Author” section, enter your conflict of interest statement in the “Confidential to Editor” section, and submit your "Accept" recommendation.

Reviewer #3: All comments have been addressed

2. Is the manuscript technically sound, and do the data support the conclusions?

Reviewer #3: Yes

3. Has the statistical analysis been performed appropriately and rigorously? 

Reviewer #3: Yes

4. Have the authors made all data underlying the findings in their manuscript fully available?

Reviewer #3: Yes

5. Is the manuscript presented in an intelligible fashion and written in standard English?

Reviewer #3: Yes

6. Review Comments to the Author

Reviewer #3: Thank you for the opportunity to re-review your work.

In this revision of the original submission, the authors appropriately addressed all comments and suggestions by the reviewers.

7. PLOS authors have the option to publish the peer review history of their article (what does this mean?). If published, this will include your full peer review and any attached files.

Reviewer #3: No

---

## [Editor Report · Acceptance letter]

2 Aug 2021

PONE-D-21-08904R1 

Evaluation of Automated Specialty Palliative Care in the Intensive Care Unit: A Retrospective Cohort Study 

Dear Dr. Secunda:

I'm pleased to inform you that your manuscript has been deemed suitable for publication in PLOS ONE. Congratulations! Your manuscript is now with our production department. 

Kind regards, 

on behalf of

Dr. Tai-Heng Chen 

Academic Editor

PLOS ONE